## [Decision Letter · Decision Letter 0]

Dear Dr. Miller,

Thank you for submitting your manuscript to PLOS ONE. After careful consideration, we feel that it has merit but does not fully meet PLOS ONE’s publication criteria as it currently stands. Therefore, we invite you to submit a revised version of the manuscript that addresses the points raised during the review process.



We look forward to receiving your revised manuscript.

Kind regards,

Johanna Pruller, Ph.D.

Associate Editor

PLOS ONE

 [Funded by an investigator-initiated grant to FAM from the Canadian Institutes of Health Research (PJT 148805).]. 

Additional Editor Comments (if provided):

Reviewers' comments:

Reviewer's Responses to Questions

**Comments to the Author**

1. Is the manuscript technically sound, and do the data support the conclusions?

Reviewer #1: Yes

Reviewer #2: Yes

2. Has the statistical analysis been performed appropriately and rigorously?

Reviewer #1: N/A

Reviewer #2: N/A

3. Have the authors made all data underlying the findings in their manuscript fully available?

Reviewer #1: No

Reviewer #2: No

4. Is the manuscript presented in an intelligible fashion and written in standard English?

Reviewer #1: Yes

Reviewer #2: Yes

Reviewer #1: The manuscript "Prognosis, treatment decision-making and value: a qualitative exploration of the emerging role of breast cancer prognostic assays" touches a very important and delicate topic, which has ramifications into multiple aspects of breast cancer post-diagnosis.

The manuscript is clearly written and the authors explore with incisiveness multiple aspects evolving around this novel technology.

Despite the fact that the value of reducing overtreatment (in this particular case, chemotherapy use) is extremely high, the authors give us a taste of how complex it is to practically introduce protocols to make it a reality for cancer patients. It emerges that two main approaches divide the world about the use of chemotherapy and the interview exerts in the manuscript give us a clear picture of the contrasting positions.

Costs, jurisdictional differences, acceptance of prognostic assays are also discussed to considerable details.

Despite the limitations of this study (which are frankly stated by the authors), I believe this topic is of utmost importance for the future of population health services. Precision medicine is about increasing the quality of life of people while taking care of the same diseases in a better way.

I really enjoyed reading this article and hope it contributes to facilitate discussion (and hopefully integrations) of those promising novel technologies, as they show the potential for a better living.

Reviewer #2: Thank you for the opportunity to review this manuscript, which explores clinician and researcher perspectives on the value, clinical utility, and integration of breast cancer prognostic assays into clinical care. The authors’ findings provide an interesting view of the current role of these assays in breast cancer treatment as perceived by clinicians. I believe that several key aspects of the study require clarification or expansion:

Methods

- Some participants were family physicians/general practitioners, who are not typically involved in requesting, processing, and interpreting breast cancer prognostic assays. What was the rationale for including these participants?

- Information on participants’ practice specialty should also be included in Table 1.

- The questions explored in the semi-structured interviews should be described in more detail. Ideally, the authors should provide the initial interview guide as part of the supplementary material.

Results

- Although most participants discussed prognostic assays supported by evidence for a predictive role (Oncotype Dx and DCISionRT), some mentioned assays lacking strong evidence for predicting treatment benefit in addition to their prognostic role. Did the authors observe any differences in perspectives based on the specific assay discussed? Were there any themes related to clinician preference for tools with both prognostic and predictive value, especially in settings where more than one assay is available?

Discussion

- According to the results, the primary perceived role of breast cancer prognostic assays is to reduce overtreatment. How does this relate to reimbursement decisions in some jurisdictions for patients with smaller tumors (e.g. T1b, node-negative cancers), since these patients are much less likely to receive chemotherapy in the absence of a prognostic assay? Were there any discussions about the use of these assays to escalate treatment for some patients?

- The authors’ findings could also inform the development of strategies to increase the value of prognostic/predictive assays, such as cost reduction, or by selecting patient subgroups for whom the clinical utility of avoiding overtreatment is higher. For instance, depending on baseline local chemotherapy utilization practices, this may apply for postmenopausal patients, or for those with node-positive disease.

- Could the authors clarify why conducting interviews virtually is considered a limitation in this study?

- In addition to the lack of patient perspectives, the absence of payer perspectives may also represent a limitation, given the manuscript’s strong focus on value, cost, and reimbursement decisions.

**Do you want your identity to be public for this peer review?** For information about this choice, including consent withdrawal, please see our Privacy Policy

Reviewer #1: No

Reviewer #2: No

---

## [Author Response · Author response to Decision Letter 1]

22 Apr 2025

Dear Dr. Chenette,

Thank you for the invitation to revise and resubmit our manuscript. We have addressed the Editor’s and Reviewer’s comments and believe that incorporating their feedback has enhanced the quality of our paper. Their comments were insightful, and we believe this review process has resulted in an improved manuscript. A detailed response to the comments may be found in the table below.

Sincerely,

Gillian Parker

Editor Comment

Action

Please ensure that your manuscript meets PLOS ONE's style requirements, including those for file naming

We have:

-Updated the title page formatting as per the provided title page guidance document

-Renamed the Supplementary file to: S1 – SRQR Reporting Checklist

Please state what role the funders took in the study.

We have added the following sentence in the funding statement:

Please confirm at this time whether or not your submission contains all raw data required to replicate the results of your study.

We have added the following data availability statement:

Data Availability Statement: The raw data for this study are transcripts of interviews, which contain potentially identifying participant information. Relevant, de-identified excerpts of the transcripts are included in the paper. For external requests or inquiries regarding data access, please contact the corresponding author. Requests will be reviewed to ensure they meet ethical and data governance guidelines.

Reviewer Comment

Action

Methods

Some participants were family physicians/general practitioners, who are not typically involved in requesting, processing, and interpreting breast cancer prognostic assays. What was the rationale for including these participants?

Thank you for highlighting this. We have corrected the role, added counts for the participants, and updated references in the manuscript.

Information on participants’ practice specialty should also be included in Table 1. Thank you for this suggestion. We have added this information to the results section. It is disaggregated from Table 1 to support anonymity.

The questions explored in the semi-structured interviews should be described in more detail. Detail has been added to the data collection section of the manuscript. In addition, the interview guide has been added as Supplementary File 2.

Ideally, the authors should provide the initial interview guide as part of the supplementary material.

We have added the Interview Guide as Supplementary File 2.

Results

Although most participants discussed prognostic assays supported by evidence for a predictive role (Oncotype Dx and DCISionRT), some mentioned assays lacking strong evidence for predicting treatment benefit in addition to their prognostic role. Did the authors observe any differences in perspectives based on the specific assay discussed?

Were there any themes related to clinician preference for tools with both prognostic and predictive value, especially in settings where more than one assay is available?

Thank you for these thoughtful questions. They offer important inquiry for future research!

We did not observe any significant or reportable themes regarding differences in perspectives based on the specific assay discussed.

The topic of clinician preference for specific tools was not included in the interview questions and we did not observe these themes in the data.

Discussion

According to the results, the primary perceived role of breast cancer prognostic assays is to reduce overtreatment. How does this relate to reimbursement decisions in some jurisdictions for patients with smaller tumors (e.g. T1b, node-negative cancers), since these patients are much less likely to receive chemotherapy in the absence of a prognostic assay?

Were there any discussions about the use of these assays to escalate treatment for some patients? As stated above, we appreciate these insightful questions and agree they are important topics to be explored in future research.

The theme of reducing overtreatment emerged in the interview data.

The topic of reimbursement decisions regarding smaller tumors was not included in the interview questions and we did not observe significant or reportable themes on this topic.

We did not observe themes regarding the use of assays to escalate treatment plans.

Limitations

Could the authors clarify why conducting interviews virtually is considered a limitation in this study?

In addition to the lack of patient perspectives, the absence of payer perspectives may also represent a limitation, given the manuscript’s strong focus on value, cost, and reimbursement decisions.

Thank you for this note. Upon reflection, virtual interviews were a strength as they facilitated the geographically diverse participant sample. We have removed this text.

Thank you for the suggestion. We have added lack of payer participation to the limitations.

---

## [Decision Letter · Decision Letter 1]

Prognosis, treatment decision-making and value: a qualitative exploration of the emerging role of breast cancer prognostic assays

PONE-D-24-17563R1

Dear Dr. Miller,

We’re pleased to inform you that your manuscript has been judged scientifically suitable for publication and will be formally accepted for publication once it meets all outstanding technical requirements.

Kind regards,

Dongling Wu

Academic Editor

PLOS ONE

---

## [Editor Report · Acceptance letter]

PONE-D-24-17563R1

PLOS ONE

Dear Dr. Miller,

I'm pleased to inform you that your manuscript has been deemed suitable for publication in PLOS ONE. Congratulations! Your manuscript is now being handed over to our production team.

Kind regards,

on behalf of

Dr. Dongling Wu

Academic Editor

PLOS ONE